# Multi-Platform Autobidding with and without Predictions

## Abstract

We study the problem of finding the optimal bidding strategy for an advertiser in a multi-platform auction setting. The competition on a platform is captured by a value and a cost function, mapping bidding strategies to value and cost respectively. We assume a diminishing returns property, whereby the marginal cost is increasing in value. The advertiser uses an autobidder that selects a bidding strategy for each platform, aiming to maximize total value subject to budget and return-on-spend constraint. The advertiser has no prior information and learns about the value and cost functions by querying a platform with a specific bidding strategy. Our goal is to design an algorithm that finds the optimal bidding strategy with a small number of queries.

We first present an algorithm that requires $O(m \log(mn) \log n)$ queries, where $m$ is the number of platforms and $n$ is the number of possible bidding strategies in each platform. Moreover, we adopt the learning-augmented framework and propose an algorithm that utilizes a (possibly erroneous) prediction of the optimal bidding strategy. We provide a $O(m \log(m\eta) \log \eta)$ query-complexity bound on our algorithm as a function of the prediction error $\eta$. This guarantee gracefully degrades to $O(m \log(mn) \log n)$. This achieves a "best-of-both-worlds" scenario: $O(m)$ queries when given a correct prediction, and $O(m \log(mn) \log n)$ even for an arbitrary incorrect prediction.

## CCS Concepts

• **Theory of computation** → **Design and analysis of algorithms**.

## Keywords

Autobidding, Multi-Platform Auctions, Algorithms with Predictions

**ACM Reference Format:**

Anonymous Author(s). 2024. Multi-Platform Autobidding with and without Predictions. In . ACM, New York, NY, USA, 10 pages. https://doi.org/10.1145/nnnnnnn.nnnnnnn

## 1 Introduction

Online advertisers often advertise across multiple platforms, such as Amazon, Bing, Google, Meta and TikTok, and face the challenging task of optimizing their bids across these platforms. The complexity comes not just from having to select a vector of bidding strategies (one for each platform), but also from the diversity of auctions used across platforms and the black-box nature of the detailed auction rules and the level of competition.

To deal with the complexity, advertisers are increasingly using automated bidding agents (aka autobidding) to bid on their behalf. An autobidder allows an advertiser to specify constraints like Budget and Return-on-Spend (ROS), and bids on their behalf to maximize value subject to the constraints. This has led to a lot of research interest in problems related to autobidding (see Aggarwal et al. [3] for a recent survey). In particular, the problem of designing bidding algorithms for the single platform setting is well-studied [1], including in the online learning setting (see Section 3 of the survey [3]). However, there is not much work on the problem of bidding optimally across multiple platforms (see the related work section for what is known).

In this paper, we study the problem of finding the optimal bidding strategy in the multi-platform setting. In particular, an advertiser aims to maximize her total value subject to a global budget and return-on-spend (ROS) constraint across all platforms. To capture the black-box nature of auction mechanisms and level of competition, we assume that the advertiser has no prior knowledge about the mapping from bids to auction outcomes for any platform. Instead, the advertiser interacts with a platform's auction by submitting a bid to the platform and observing the corresponding cost and value (i.e., the user has "query access" to the mapping). We propose algorithms that find the optimal bidding strategy in this setting, and prove worse-case query complexity bounds for them.

While worst-case results offer robustness and broad applicability, the guarantees they provide can sometimes be overly pessimistic. To address this, a new framework called "algorithms with predictions" has recently been introduced. This framework allows algorithms to incorporate potentially flawed machine-learned predictions as a guiding tool. The objective is twofold: to achieve improved performance guarantees when the prediction is accurate (a property known as *consistency*) and to maintain good worst-case bounds even when the prediction is completely incorrect (a property called *robustness*). This framework provides a natural way to integrate machine-learned predictions into the design of algorithms while preserving the essential robustness offered by worst-case analysis. In this work, we adopt the learning-augmented framework and explore the role of predictions in bidding strategy optimization. Specifically, we examine the scenario where the algorithm has access to a prediction $\hat{\boldsymbol{\mu}}$ of the optimal bidding strategy, without any assumption regarding the prediction's accuracy. We propose algorithms that leverage the untrusted prediction to achieve improved query complexity bounds, which degrade gracefully based on the quality of the prediction.

### 1.1 Our Results

We model the problem of searching for the optimal bidding strategy as follows: there are $m$ platforms. For each platform $j$, we are given a cost function $c_j$ and a value function $v_j$, and $n$ different bidding strategies indexed 1 through $n$ such that $c_j(\mu_j)$ and $v_j(\mu_j)$ are respectively the cost incurred by bidding $\mu_j$ on platform $j$, and the value accrued from this bidding strategy. Our goal is to find

a collection $\boldsymbol{\mu} = (\mu_1, \ldots, \mu_m)$ such that (a) the ratio of the total value accrued to the total cost is at least some target threshold, (b) the total cost is less than the budget, and (c) the total value is maximized.

We propose a search algorithm, MEDIANOFMEDIANS, that determines the exact optimal bidding strategy using $O(m \log(mn) \log n)$ queries. Our algorithm builds on a characterization of the optimal bidding strategy in the multi-platform setting, which was first developed in [4] under continuous strategy space. Intuitively, the optimal strategy is to keep increasing bid on the platform that currently offers the highest marginal bang-per-buck (corresponding to the lowest marginal cost-per-unit-value) until either the budget or the ROS constraint is about to be violated. In other words, the optimal strategy aims to equalize the marginal cost-per-unit-value across all the platforms.

At a high level, the algorithm searches in the space of marginal costs. Initially, there are up to $mn$ candidate marginal costs. The algorithm carefully selects a candidate marginal cost and finds the corresponding vector of bidding strategies, as defined in Lemma 1, via Subroutine 2. Based on the outcomes (i.e. cost and value on each platform) of the corresponding strategy vector, the algorithm removes a constant fraction of candidate marginals from consideration, and recurses on the residual problem.

We complement this algorithmic result with an $\Omega(m \log n)$ lower bound and an $\Omega(\log mn)$ lower bound for this problem. The $\Omega(\log mn)$ lower bound reflects the difficulty of identifying the optimal marginal cost among the $mn$ possible candidates, while the $\Omega(m \log n)$ bound captures the complexity of determining the corresponding bidding strategy for that optimal marginal cost. Notably, these are the two key components of our algorithm, and our upper bound is the product of these two lower bounds.

Next, we adopt the learning-augmented framework to improve the worst-case query complexity bound. We propose an algorithm with access to a prediction $\hat{\boldsymbol{\mu}}$ of the optimal strategy $\boldsymbol{\mu}^o$. The algorithm, BRANCHOUTMEDIANOFMEDIANS, starts with trying to the find the optimal solution in a small range around the predictions, and expands the search range if the search is unsuccessful. With the right sequence of expanding ranges, we show that the algorithm finds the optimal strategy $\boldsymbol{\mu}^*$ with $O(m \log(m\eta) \log \eta)$ queries, where $\eta = \max_j |\boldsymbol{\mu}_j^* - \hat{\boldsymbol{\mu}}_j|$ represents the prediction error. This means that the algorithm requires only $O(m)$ queries when the predictions are accurate; this is the minimum number of queries needed to implement any bidding strategy. Moreover, since $\eta \leq n$, the total number of queries never exceeds that of the MEDIANOFMEDIANS algorithm.

## 1.2 Related Work

*Multi-platform mechanism design and autobidding.* Previous research has examined the multi-platform auction environment from both the auctioneer's and the bidders' perspectives. Regarding auction design, Aggarwal et al. [2] analyzes a scenario where a single platform manages multiple channels, each selling queries via a second-price auction (SPA) with a reserve price. The authors assess the costs associated with each channel optimizing its own reserve price compared to a unified platform policy. Inspired by the Display Ad market, Paes Leme et al. [32] explores a model in which

multiple platforms vie for profit-maximizing bidders who must use the same bid across all platforms (which we refer to as a uniform bid). Their key finding indicates that the first-price auction (FPA) serves as the optimal auction format for these platforms. On the bidders' side, Susan et al. [34] investigates bidding strategies for utility-maximizing advertisers operating across multiple platforms while adhering to budget constraints. Meanwhile, Deng et al. [20] focuses on value-maximizing advertisers and reveals that optimizing ROS per platform can yield arbitrarily poor results when both ROS and budget constraints are in play.

Aggarwal et al. [4] study a similar multi-platform setting with autobidders under ROI constraints but focus on the auction design problem from the platform's perspective. While first-price auctions are optimal in the absence of competition (Deng et al. [19]), they show that from the perspective of each separate platform, running a second-price auction can achieve larger revenue than first-price auction when there are two competing platforms. They also identify key factors influencing the platform's choice of auction formats, including advertiser sensitivity to auction changes, competition and relative inefficiency of second-price auctions. In our paper, we focus on how advertisers can bid optimally in the multi-platform setting.

*Algorithms with predictions.* In recent years, the learning-augmented framework has emerged as a prominent paradigm for the design and analysis of algorithms. For an overview of early contributions, we refer to [31], while [27] offers an up-to-date compilation of relevant papers in this area. This framework seeks to address the shortcomings of overly pessimistic worst-case analyses. In the last five years alone, hundreds of papers have explored traditional algorithmic challenges through this lens, with notable examples including online paging [29], scheduling [33], optimization problems related to covering [12] and knapsack constraints [24], as well as topics like Nash social welfare maximization [13], the secretary problem [6, 21, 22], and various graph-related problems [7].

More closely related to our work, the research on learning-augmented mechanisms interacting with strategic agents is recently initiated by Agrawal et al. [5] and Xu and Lu [35]. This area includes strategic facility location [5, 14, 25, 35], strategic scheduling [10, 35], auctions [11, 16, 28, 30, 35], bicriteria mechanism design (which seeks to optimize both social welfare and revenue) [8], graph problems with private input [18], metric distortion [15], and equilibrium analysis [23, 26]. Recently, Christodoulou et al. [17] revisited mechanism design challenges by focusing on predictions about the outcome space rather than the input. While most of these studies concentrate on the mechanism design problem, our research emphasizes how predictions can assist agents in identifying optimal strategies. For more information on this body of work, we recommend [9].

## 2 Preliminaries

We consider the problem of finding the optimal bidding strategy in a multi-platform auction setting for a value-maximizer with *budget* and *return on spend* (ROS) constraints. There is a set $M$ consisting of $m$ platforms in the market. We assume that for each platform, the advertiser can pick from $n$ different bids (note that this set can be different for different platforms), indexed by $0, 1, 2, \ldots, n$, where

bid 0 is used to denote non-participation. Each platform $j \in M$ is described by a value and a cost function that map bid indices to a corresponding value and cost, respectively, i.e., $v_j : \{0, 1, \ldots, n\} \to \mathbb{R}^{\geq 0}$ and $c_j : \{0, 1, \ldots, n\} \to \mathbb{R}^{\geq 0}$. In other words, when a bidder chooses to bid according to bid $\mu \in \{0, 1, \ldots, n\}$, they incur a cost of $c_j(\mu)$ and receive a value of $v_j(\mu)$. We assume that there is a strict ordering of costs and values by bid index, i.e. $v_j(\mu) < v_j(\mu + 1)$ and $c_j(\mu) < c_j(\mu + 1)$ for all $\mu \in \{0, 1, \ldots, n\}$. We refer to the mapping from bid to the cost and value of each platform as the *landscape* of that platform. In addition, we define the *marginal cost* of bidding $\mu \geq 1$ on platform $j$ as

$$\mathsf{MC}_j(\mu) = \frac{c_j(\mu) - c_j(\mu - 1)}{v_j(\mu) - v_j(\mu - 1)}, \qquad \text{(Marginal Cost)}$$

where $c(0) = 0$ and $v(0) = 0$. We make standard convexity assumption that $\mathsf{MC}_j$ is non-decreasing for every platform $j$.

Given the integral strategy set, we expand the bidding space by also considering the fractional solution between each integral bid, hence making the strategy space continuous. We use $S$ and $S^c$ to denote the integral and fractional strategy space, respectively. The cost, value and marginal functions of the continuous bidding space $[0, n]^m$ extend the discrete function by linear interpolation.[1] Formally

$$v_j(\mu) = (\lceil \mu \rceil - \mu) \cdot v_j(\lfloor \mu \rfloor) + (\mu - \lfloor \mu \rfloor) \cdot v_j(\lceil \mu \rceil),$$

$$c_j(\mu) = (\lceil \mu \rceil - \mu) \cdot c_j(\lfloor \mu \rfloor) + (\mu - \lfloor \mu \rfloor) \cdot c_j(\lceil \mu \rceil),$$

Consequently, we have that $\mathsf{MC}_j(\mu) = \mathsf{MC}_j(\lceil \mu \rceil)$.

We note that the problem of finding the optimal integral solution is NP-hard.[2] The objective of the bidder is therefore to find an optimal *fractional* bidding strategy $\boldsymbol{\mu} = (\mu_1, \mu_2, \ldots, \mu_m)$ where $\mu_j \in [0, n]$ such that she maximizes the total value received by executing bidding strategy $\mu_j$ on each platform $j$, subject to a budget constraint and the ROS constraint *across all platforms*. Let $B$ and $T$ be the budget and target ROS of the bidder. We can formulate the problem as the following program:

$$\max_{\boldsymbol{\mu} = (\mu_1, \mu_2 \ldots, \mu_m)} \sum_{j \in M} v_j(\mu_j)$$

$$s.t. \sum_{j \in M} c_j(\mu_j) \leq T \cdot \sum_{j \in M} v_j(\mu_j), \qquad (1)$$

$$\sum_{j \in M} c_j(\mu_j) \leq B.$$

throughout the paper, we denote $\boldsymbol{\mu}^o$ the optimal (fractional) bidding strategy, and $\boldsymbol{\mu}^*$ the floor of it, i.e., $\mu_j^* = \lfloor \mu_j^o \rfloor$ for all platform $j$. Note that $\mu_j^* \in S$.

We assume that the bidder only knows the set of possible bidding strategies, but has no information about the platforms' value and cost functions. Instead, the bidder can interact with platforms via *bidding queries*: the bidder plays strategy $\mu$ on a platform $j$ to learn the value $v_j(\mu)$, the cost $c_j(\mu)$, and the marginal cost $\mathsf{MC}_j(\mu)$[3]. Each

such query is costly to the bidder, and the goal is to minimize the number of queries required to determine the optimal strategy.

Given an instance $\mathcal{I}$ and an algorithm ALG, let $\mathsf{ALG}(\mathcal{I})$ denote the number of queries needed to find the optimal strategy for that instance. Then the query complexity of the algorithm is defined as:

$$\max_{\mathcal{I}} \mathsf{ALG}(\mathcal{I})$$

*The learning-augmented framework.* In this work, we adopt the learning-augmented framework and study how we can further reduce the query complexity by considering search algorithms that are equipped with a (potentially erroneous) prediction $\hat{\boldsymbol{\mu}} \in [0, n]^m$ of the optimal fractional bidding strategy $\boldsymbol{\mu}^o(\mathcal{I}) = (\mu_1^o, \mu_2^o, \ldots, \mu_n^o)$. The *error* of an predictions $\eta$ is defined to be the maximum pointwise deviation from $\boldsymbol{\mu}^o$, formally:

$$\eta(\hat{\boldsymbol{\mu}}, \mathcal{I}) = \max_j |\hat{\mu}_j - \mu_j^o(\mathcal{I})|$$

We let the algorithm ALG use both the instance $\mathcal{I}$ and the prediction $\hat{\boldsymbol{\mu}}$ as input. We evaluate the performance of such an algorithm using its *robustness*, *consistency* and the query complexity as a function of the prediction error.

The robustness of an algorithm refers to the worst-case query complexity of the algorithm given an adversarially chosen, possibly erroneous, prediction. Mathematically,

$$\text{robustness}(\mathsf{ALG}) = \max_{\hat{\boldsymbol{\mu}}, \mathcal{I}} \mathsf{ALG}(\hat{\boldsymbol{\mu}}, \mathcal{I})$$

The consistency of an algorithm refers to the worst-case query complexity when the prediction that it is provided with is accurate, i.e., $\hat{\boldsymbol{\mu}} = \boldsymbol{\mu}^*(\mathcal{I})$. Mathematically,

$$\text{consistency}(\mathsf{ALG}) = \max_{\hat{\boldsymbol{\mu}}, \mathcal{I}: \hat{\boldsymbol{\mu}} = \boldsymbol{\mu}^*(\mathcal{I})} \mathsf{ALG}(\hat{\boldsymbol{\mu}}, \mathcal{I}).$$

Lastly, the query complexity of an algorithm given a prediction with error $\eta'$ is defined to be:

$$\max_{\hat{\boldsymbol{\mu}}, \mathcal{I}: \eta(\hat{\boldsymbol{\mu}}, \mathcal{I}) \leq \eta'} \mathsf{ALG}(\hat{\boldsymbol{\mu}}, \mathcal{I}).$$

## 3 Characterization of Bidder's Optimal Bidding Strategy

In this section, we present a characterization of the optimal bidding strategy $\boldsymbol{\mu}^o$ that will be useful in designing the algorithm. To this end, we first prove a useful lemma about the "ranking" of integral strategies in $S$. We then argue how an "almost-optimal" integral solution can be used to determine the optimal fractional solution.

LEMMA 1. *Given some positive number $k$, and the $n$ discrete indices on each platform, define $\boldsymbol{\mu}^k = (\mu_1^k, \mu_2^k, \ldots \mu_m^k)$ where*

$$\mu_j^k = \underset{\mu \in \{0, 1, \ldots, n\}}{\arg \max} \{\mathsf{MC}_j(\mu) \leq k\},$$

*then there exist a $k^*$ such that for any $k \leq k^*$, $\boldsymbol{\mu}^k$ is a feasible solution for program(1) and for any $k' > k^*$, $\boldsymbol{\mu}^{k'}$ is not feasible.*

We first show the following helper lemma. Intuitively, if we consider the landscape of each platform, and connect each bidding strategy with a straight line, the landscape would be convex, the lemma below is simply a property of a convex function. Due to space limitations, we defer the proof to appendix A.

---

[1]It can be viewed as bidding randomly between two adjacent bids.
[2]It is not hard to see that we can encode any knapsack problem as an instance of our problem with a budget constraint.
[3]When marginal cost is not part of the query output, it is still achievable by querying both the current and the previous bid, which increases the query complexity by a constant factor.

**Lemma 2.** *For any platform $j \in M$, $\frac{c_j(\mu)}{v_j(\mu)} \leq \mathrm{MC}_j(\mu)$.*

**Proof of Lemma 1.** let $k$ be the smallest $k$ with infeasible $\boldsymbol{\mu}^k$, if the infeasibility is due to the budget constraint, then for any $k' \geq k$ we trivially have that $\boldsymbol{\mu}^{k'}$ violates the budget constraint as well since $\mu_j^{k'} \geq \mu_j^k$ and the cost functions are monotone.

If the infeasibility is due to the ROS constraint, i.e.,

$$\sum_{j \in M} c_j(\mu_j^k) > T \cdot \sum_{j \in M} v_j(\mu_j^k), \tag{2}$$

proving the statement is equivalent to proving for any $k' \geq k$, $\boldsymbol{\mu}^{k'}$ is also infeasible. To this end, we first show that the maximum marginals among the $\mu_j^k$ is strictly more than $T$, assume for contradiction, that $\mathrm{MC}_j(\mu_j^k) \leq T$ for all $j$, by lemma 2 we would have the $c_j(\mu_j^k)/v_j(\mu_j^k) \leq \mathrm{MC}_j(\mu_j^k) \leq T$, which contradicts with (2). We therefore have

$$\max_{j \in M} \mathrm{MC}_j(\mu_j^k) > T \tag{3}$$

We now inductively prove that for any $k' \geq k$, we have $\boldsymbol{\mu}^{k'}$ is infeasible. Consider increasing $k'$ starting from $k$, at the beginning we could have $\boldsymbol{\mu}^{k'} = \boldsymbol{\mu}^k$ (which is infeasible), consider the first point $k' \geq k$ such that $\boldsymbol{\mu}^{k'} \neq \boldsymbol{\mu}^k$, we know that:

(1) there exist at least one platform $j'$ such that $\mu_{j'}^{k'} = \mu_{j'}^k + 1$

(2) $\mathrm{MC}_{j'}(\mu_{j'}^{k'}) > \max_{j \in M} \mathrm{MC}_j(\mu_j^k) > T$,

where the first inequality is by definition of $\boldsymbol{\mu}^{k'}$ and the second inequality is due to (3). Now consider:

$$\sum_{j \in M} c_j(\mu_j^{k'}) = \sum_{j \in M} c_j(\mu_j^k) + c_{j'}(\mu_{j'}^k + 1) - c_{j'}(\mu_{j'}^k)$$

$$> T \cdot \sum_{j \in M} v_j(\mu_j^k) + c_{j'}(\mu_{j'}^k + 1) - c_{j'}(\mu_{j'}^k)$$

$$> T \cdot \sum_{j \in M} v_j(\mu_j^k) + \mathrm{MC}_{j'}(\mu_{j'}^k + 1) \cdot (v_{j'}(\mu_{j'}^k + 1) - v_{j'}(\mu_{j'}^k))$$

$$> T \cdot \sum_{j \in M} v_j(\mu_j^k) + T \cdot (v_{j'}(\mu_{j'}^k + 1) - v_{j'}(\mu_{j'}^k))$$

$$> T \cdot \sum_{j \in M} v_j(\mu_j^{k'})$$

Inductively apply this argument for each update of $\boldsymbol{\mu}^{k'}$ proves the statement. $\square$

## 3.1 The fractional optimal bidding strategy

We present the optimal fractional solution, which at a high level is achieved by the following greedy process: starting with an initial budget of $B$, we allocate an infinitesimal amount to the platform currently offering the best value-to-unit-cost ratio (equivalently, the smallest marginal cost). We continue this process until either the budget is exhausted or the Return on Spend (ROS) constraint becomes tight. The bids corresponding to the final cost/value ratio on each platform form the optimal fractional strategy.

For ease reference, we formally define the feasible $\boldsymbol{\mu}^k$ with the largest $k$ as the "almost-optimal" integral solution.

**Definition 1 (Almost-Optimal Strategy).** *We say an bidding strategy $\boldsymbol{\mu}$ is almost-optimal if $\boldsymbol{\mu} = \max_k \left[ \boldsymbol{\mu}^k \text{ is feasible} \right]$.*

Essentially, the almost-optimal integral strategy provides a close lower bound for the optimal fractional solution $\boldsymbol{\mu}^o$. Specifically, let $\boldsymbol{\mu}^*$ represent the largest feasible $\boldsymbol{\mu}^k$; then, $\boldsymbol{\mu}^* = \lfloor \boldsymbol{\mu}^o \rfloor$. We present a subroutine that takes the almost-optimal integral solution $\boldsymbol{\mu}^*$ as input and returns the exact fractional optimal bidding strategy $\boldsymbol{\mu}^o$ by greedily selecting the smallest marginal costs (breaking ties lexicographically with respect to a fixed ordering of platforms) until the constraint is tight.

Consider the following equation:

$$\sum_j (x_j \cdot c_j(\mu_j^*) + (1 - x_j) \cdot c_j(\mu_j^* + 1))$$

$$= \min \left[ B, T \cdot \left( \sum_j (x_j \cdot v_j(\mu_j^*) + (1 - x_j) \cdot v_j(\mu*_j + 1)) \right) \right] \tag{4}$$

By definition of $\boldsymbol{\mu}^*$, we have the $x_j \in [0, 1]$ for all platform $j$. In the case where there are multiple set of solutions, we break ties by maximizing the $x_j$ with lower platform index first.

---

**SUBROUTINE 1:** RoundUp

**Input:** almost-optimal integral solution $\boldsymbol{\mu}^*$

1 **for** $j \in M$ **do**
2      $\mu_j' \leftarrow \mu_j^* + 1$
3      $\mu_j^o \leftarrow \mu_j^*$
4      query each $\mu_j'$ to obtain $v_j(\mu_j'), c_j(\mu_j')$ and $\mathrm{MC}_j(\mu_j')$
5 re-index the platforms in non-decreasing order of $\mathrm{MC}_j(\mu_j')$
     s.t. if $i \leq j$, $\mathrm{MC}_i(\mu_i') \leq \mathrm{MC}_j(\mu_j')$
6 Solve for $x_j$ in (4)
7 $\mu_j^o \leftarrow \mu_j^o + x_j$
8 **return** $\boldsymbol{\mu}^o$

---

**Lemma 3 (Optimal Bidding Strategy).** *Let $\boldsymbol{\mu}^*$ be the almost-optimal integral solution. Then RoundUp $(\boldsymbol{\mu}^*)$ is bidder's fractional optimal bidding strategy.*

**Proof.** We prove optimality of $\boldsymbol{\mu}' = $ RoundUp $(\boldsymbol{\mu}^*)$ by contradiction. Assume there exist another solution $\boldsymbol{\mu}^a$ that is optimal with a value strictly higher than $\boldsymbol{\mu}'$. If $\boldsymbol{\mu}^a$ obtains a better value than $\boldsymbol{\mu}'$, there must be at least one platform $j$ such that $\mu_j^a > \mu_j'$. Consequently, there must be some other platform $i$, such that $\mu_j^a < \mu_j'$, since the constraints are tight of solution $\boldsymbol{\mu}'$ by definition of the RoundUp algorithm and $\boldsymbol{\mu}^a$ is feasible by assumption.

we first argue that $\mathrm{MC}_j(\mu_j^a) > \mathrm{MC}_i(\mu_i')$. To see this, first note that $\mathrm{MC}_j(\mu_j^a) \geq \mathrm{MC}_i(\lfloor \mu_i' \rfloor)$, since otherwise by definition of $\boldsymbol{\mu}^k$ we have:

$$\mu_j' = \arg\max_\mu [\mathrm{MC}_j(\mu) \leq k] \geq \arg\max_\mu [\mathrm{MC}_j(\mu) \leq \mathrm{MC}_i(\lfloor \mu_i' \rfloor)] \geq \lceil \mu_j^a \rceil,$$

where the last inequality is since $\mathrm{MC}(\mu) = \mathrm{MC}(\lceil \mu \rceil)$ for any $\mu$, this contradicting with the assumption that $\mu_j' < \mu_j^a$. By the greedy

natural, with a similar contradiction argument we can show that $MC_j(\mu_j^a) > MC_i(\mu_i')$, we then have:

$$MC_j(\mu_j^a) > MC_i(\mu_i') > MC_i(\mu_i^a), \tag{5}$$

where the last inequality is due to the assumption that $\mu_i' \geq \mu_i^a$ and the monotonicity of $MC$ functions.

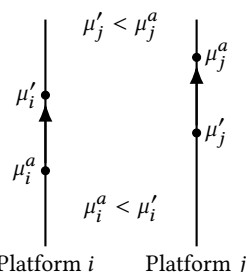

Platform $i$      Platform $j$

Now we argue that $\boldsymbol{\mu}^a$ can be further improved by an exchange argument, contradicting with the assumption that $\boldsymbol{\mu}^a$ is optimal. Consider again the bidding strategy $\boldsymbol{\mu}^a$, consider reduce $\mu_j^a$ by some $\epsilon$ amount, and increase on $\mu_i^a$ by the corresponding amount until the constraints are tight again, since $MC_j(\mu_j^a) \geq MC_i(\mu_i^a)$, the "bang per buck" for the exchange portion strictly increases, contradicting with the assumption that $\boldsymbol{\mu}^a$ is optimal. □

## 4 The Median of the Medians Algorithm

In this section, with the help of the characterization in the previous section, we present an algorithm with a worst-case query complexity of $O(m \log(mn) \log n)$. Note that if our feasible region were downward-closed, there would be a straightforward algorithm to solve the problem: We could perform a high-dimensional binary search in the bidding space, cutting down the whole strategy space by a constant fraction each time we query a particular strategy. This would lead to $m \cdot \log(n^m) = m^2 \log n$ queries (since querying one vector of strategies requires submitting a bidding strategy on each of the $m$ platforms). Unfortunately, in the example below we show that our feasible region is not necessarily downward-closed.

*Example 1.* Consider a simple example with two platforms, 1 and 2, both having the following cost and value functions:

$$c_1(\mu) = \mu \quad v_1(\mu) = \frac{8}{3}\mu;$$

$$c_2(\mu) = \mu \quad v_2(\mu) = \mu.$$

Suppose the constraints are $B = 10$ and $T = \frac{1}{2}$. First, observe that $\mu_1 = \frac{3}{2}$ and $\mu_2 = 1$ is a feasible solution since:

$$2 \cdot \left(\frac{3}{2} + 1\right) = \frac{8}{3} \cdot \frac{3}{2} + 1.$$

However, if we reduce $\mu_1$ to 1, we get:

$$2 \cdot (1 + 1) > \frac{8}{3} + 1,$$

indicating that the updated bidding strategy is no longer feasible. Therefore, the feasible region is not downward-closed.

Without a downward-closed feasible region, it is unclear which bidding strategy to try or how the search algorithm should proceed to minimize the number of attempts. To address this, we leverage the structure of integral bidding strategies shown in previous section and focus on identifying the $k$ values corresponding to the maximum feasible $\boldsymbol{\mu}^k$ first. The potential $k$ values are the set of marginal costs across all platforms (there are $mn$ of them). We utilize the "median of medians" idea to ensure that we eliminate a constant fraction of marginal cost options with each round of probing.

First, given the characterization of the optimal solution, we provide two subroutines that are useful for our algorithm. We first provide a subroutine named MatchingMC, that given a $k$, finds the $\boldsymbol{\mu}^k$ vector via binary search on each platform. We show that the query complexity of this subroutine is $O(m \log n)$. Whenever we call the subroutine, we would make sure that the $MC_j(\ell_j) \leq k$, i.e., there is at least one strategy in the search range that is feasible. Due to space limitations, we defer the following two proofs to appendix B

---

**SUBROUTINE 2:** MatchingMC

**Input:** search range $[\ell_j, r_j]$ of each $j$, the target MC $k$

1   **for** $j \in M$ **do**
2      **while** $\mu_j^k = $ NULL **do**
3          $\mu_j \leftarrow \frac{\ell_j + r_j}{2}$ for all $j \in M$    // Binary search on each platform
4          **if** $MC_j(\mu_j) \leq k$ **then** $\ell_j \leftarrow \mu_j$
5          **if** $MC_j(\mu_j) > k$ **then** $r_j \leftarrow \mu_j - 1$
6          **if** $r_j \leq \ell_j$ **then** $\mu_j^k \leftarrow \ell_j$
7   **return** $\boldsymbol{\mu}^k = (\mu_1^k, \mu_2^k, \ldots, \mu_m^k)$

---

LEMMA 4. *Given some $k \geq 0$, MatchingMC outputs the corresponding $\boldsymbol{\mu}^k$ with at most $O(m \log \max_j (r_j - \ell_j))$ queries.*

We now provide a subroutine that check if a given integral bidding profile $\boldsymbol{\mu}$ is the almost-optimal solution (defined in Definition 1) or not. In addition, the subroutine can also check if a bidding profile is in the form of $\boldsymbol{\mu}^k$ for some $k$ (defined in Lemma 1). The worst-case query complexity is $O(m)$.

LEMMA 5. *Given a bidding profile $\boldsymbol{\mu}$, the subroutine OptCheck determines if the given $\boldsymbol{\mu}$ is* INFEASIBLE, NOT $\boldsymbol{\mu}^k$, NOT-OPTIMAL *or* ALMOST-OPTIMAL *with at most $O(m)$ queries.*

We are now ready to present our algorithm, MedianOfMedians. This algorithm finds the *almost-optimal integral solution* by searching within the marginal cost space and then converts this almost-optimal integral solution to the optimal fractional solution using the RoundUp procedure. The search process is inspired by the median-of-medians algorithm. In each iteration, we first identify the median marginal cost for each platform, and then select the median that most evenly splits the space, i.e., ensuring that the number of marginals weakly smaller than this median is equal to the number of marginals weakly larger than it.

Next, we use MatchingMC to determine the corresponding bidding profile $\boldsymbol{\mu}^k$ with the median-of-the-medians marginal as the $k$-value, and apply OptCheck to evaluate the quality of the

**SUBROUTINE 3:** OPTCHECK

**Input:** some bidding strategy $\boldsymbol{\mu}$

1 query each platform $j$ strategy $\mu_j$, obtain $v_j(\mu_j)$, $c_j(\mu_j)$ and $\text{MC}_j(\mu_j)$

2 **if** $\boldsymbol{\mu}$ is infeasible **then return** INFEASIBLE

3 $\bar{j} \leftarrow \arg\max_j[\text{MC}_j(\mu_j)]$

4 $k \leftarrow \text{MC}_{\bar{j}}(\mu_{\bar{j}})$

5 **for** $j \in M$ **do**

6      $\mu'_j \leftarrow \mu_j + 1$     `// Check the next MC value for platform j`

7      query $\mu'_j$ on platform $j$ obtain $v_j(\mu'_j)$, $c_j(\mu'_j)$ and $\text{MC}_j(\mu'_j)$

8      **if** $\text{MC}_j(\mu'_j) \leq k$ for any $j \neq \bar{j}$ **then**

9          **return** NOT-$\boldsymbol{\mu}^k$       `// μ is not μᵏ for some k`

10 $j^* \leftarrow \arg\min_j(\text{MC}_j(\mu'_j))$    `// find the minimum among the next points`

11 $\mu_{j^*} \leftarrow \mu'_{j^*}$

12 **if** $\boldsymbol{\mu}$ is infeasible       `// the updated μ is not feasible`

13 **then return** ALMOST-OPTIMAL

14 **else return** NOT-OPTIMAL

bidding profile $\boldsymbol{\mu}^k$. Based on the result from OPTCHECK($\boldsymbol{\mu}^k$), we can eliminate a constant fraction of the remaining candidates for the optimal marginal costs. This process is repeated iteratively until we find an almost-optimal solution. Finally, we apply the ROUNDUP procedure to obtain the fractional optimal solution. For a formal description, please refer to Algorithm 1.

**ALGORITHM 1:** MEDIANOFMEDIANS

1 **Initialize:** $\ell_j \leftarrow 1$, $r_j \leftarrow n$ for all $j \in M$

2 **while** TRUE **do**

3      $\mu_j \leftarrow \frac{\ell_j + r_j}{2}$ for all $j \in M$

4      query each platform $j$ strategy $\mu_j$, obtain $v_j(\mu_j)$, $c_j(\mu_j)$ and $\text{MC}_j(\mu_j)$

5      rank the platforms in non-decreasing order of $\mu_j$ s.t. if $i \leq j, \mu_i \leq \mu_j$

6      $j^* \leftarrow \min_j(|\sum_{i \leq j}(r_i - \ell_i) - \sum_{i \geq j}(r_i - \ell_i)|)$   `// find the j* that equally split the search space`

7      $\boldsymbol{\mu}^* \leftarrow$ MATCHINGMC($[1, n]$ for all $j$, $\text{MC}_{j^*}(\mu_{j^*})$)

8      **if** OPTCHECK($\boldsymbol{\mu}^*$) = INFEASIBLE **then**

9          $r_j \leftarrow \mu_j - 1$ for all $j \geq j^*$   `// reduce the search space`

10      **else if** OPTCHECK($\boldsymbol{\mu}^*$) = NOT-OPTIMAL **then**

11          $\ell_j \leftarrow \mu_j + 1$ for all $j \leq j^*$

12      **else if** OPTCHECK($\boldsymbol{\mu}^*$) = ALMOST-OPTIMAL **then**

13          **return** ROUNDUP($\boldsymbol{\mu}^*$)

In the rest of the section, we prove the correctness and query complexity of the algorithm.

THEOREM 2. *Given any instance $\mathcal{I}$, the MEDIANOFMEDIANS algorithm finds the fractional optimal bidding strategy with at most $O(m \log mn \log n)$ queries.*

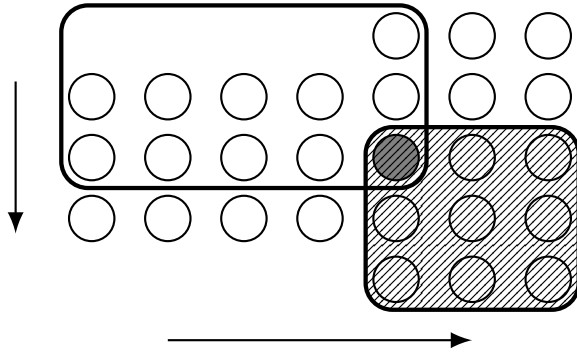

increasing median marginal cost

**Figure 1: Illustration of one round of MEDIANOFMEDIANS. Each column represents the current search region of a platform. The vertical arrow indicates the increasing direction of $\mu$ in each platform. The platforms are ranked by the median marginals as described in the algorithm. The grey point represents the queried $k$ value. If $\boldsymbol{\mu}^k$ is infeasible, all strategies in the shaded round rectangle are removed; otherwise, all strategies in the non-shaded one are removed.**

PROOF. We first prove the correctness of the algorithm. By Lemma 3 we know that the almost-optimal integral bidding strategy correspond to $\boldsymbol{\mu}^{k^*}$ where $k^*$ is the maximum $k$ such that $\boldsymbol{\mu}^k$ is feasible. We prove the correctness of the algorithm by first showing that during the execution of the algorithm, there always exist some $\mu \in [\ell_j, r_j]$, of which the $\boldsymbol{\mu}^{\text{MC}_j(\mu)} = \boldsymbol{\mu}^{k^*}$. In other words, the algorithm can not eliminate the critical marginal cost $\text{MC}_j(\mu)$ that corresponds to $\boldsymbol{\mu}^{k^*}$. Consider the possible updates of $\ell_j$ and $r_j$ for each platform $j$, i.e., Line 9 and Line 11. First consider any iteration such that OPTCHECK($\boldsymbol{\mu}^*$) = INFEASIBLE, and for all platforms $j \geq j^*$ w.r.t to the ranking defined in Line 5, we have $\text{MC}_j(\mu_j) \geq \text{MC}_{j^*}(\mu_{j^*})$. By monotonicity, for any $\mu \geq \mu_j$ on platform $j$ we have:

$$\text{MC}_j(\mu) \geq \text{MC}_j(\mu_j) \geq \text{MC}_{j^*}(\mu_{j^*}),$$

By Lemma 1, since OPTCHECK($\boldsymbol{\mu}^*$) = INFEASIBLE, we would also have $\boldsymbol{\mu}^{\text{MC}_j(\mu)}$ is infeasible for $\mu \geq \mu_j$ for platforms $j \geq j^*$. Therefore, Line 9 does not remove any $\mu$ of which $\boldsymbol{\mu}^{\text{MC}_j(\mu)} = \boldsymbol{\mu}^{k^*}$.

Next consider any iteration such that OPTCHECK($\boldsymbol{\mu}^*$) = NOT-OPTIMAL, and for all platforms $j \leq j^*$ w.r.t to the ranking defined in Line 5, we have $\text{MC}_j(\mu_j) \leq \text{MC}_{j^*}(\mu_{j^*})$. Again by monotonicity, for any $\mu \leq \mu_j$ on platform $j$ we have:

$$\text{MC}_j(\mu) \leq \text{MC}_j(\mu_j) \leq \text{MC}_{j^*}(\mu_{j^*}),$$

By Lemma 1, since OPTCHECK($\boldsymbol{\mu}^*$) = NOT-OPTIMAL, we have $\boldsymbol{\mu}^{\text{MC}_j(\mu)}$ is also feasible and not optimal for $\mu \leq \mu_j$ for platforms $j \leq j^*$. Therefore Line 11 does not remove any $\mu$ of which $\boldsymbol{\mu}^{\text{MC}_j(\mu)} = \boldsymbol{\mu}^{k^*}$ as well. In addition, it is easy to see that $\boldsymbol{\mu}^{\text{MC}_j(\mu)} = \boldsymbol{\mu}^{k^*}$ for some platform $j$ and some strategy $\mu$. (let $\text{MC}_j(\mu) = \arg\max(\text{MC}_j(\mu_j^{k^*}))$). And since the set of bids is finite and getting strictly smaller in each round, the algorithm will eventually terminate with the almost-optimal integral bidding solution $\boldsymbol{\mu}^{k^*}$, after which applying ROUNDUP would give us the fractional optimal solution.

We now prove the query complexity of the algorithm. In particular, we argue that the while loop would iterate no more then $O(\log(mn))$ times. Together with the $O(m \log n)$ query complexity of MatchingMC this would show that the query complexity of the algorithm is $O(m \log(mn) \log n)$. First note that there are in total $m \cdot n$ possible marginal costs ($MC_j(\mu)$ for all $j$ and $\mu$). By definition of $j^*$, and $\mu_j$ for each platform $j$, we have that $\min(|\{MC_i(\mu) : i \le j^* \text{ and } \mu \le \mu_i\}|, |\{MC_i(\mu) : i \ge j^* \text{ and } \mu \ge \mu_i\}|) \ge \frac{\sum_j (r_j - \ell_j)}{4} - \min_j (r_j - \ell_j) = O(\sum_j (r_j - \ell_j))$. Since we remove a constant fraction of choices in each round, the number of queries is no more then $O(\log mn)$. Lastly note that RoundUp makes at most $m$ queries, making the total queries needed for MedianOfMedians $O(m \log(mn) \log n)$.                                                              □

## 5  Lower Bounds on Query Complexity

In this section, we provide some lower bounds on query complexity. We show that any algorithm needs to have a query complexity of $\Omega(m \log n)$ even if it knows the optimal marginal cost $k$. We also provide a lower bound of $\Omega(\log mn)$ for finding the optimal marginal cost $k$ even when the algorithm is given a single-query black-box oracle for MatchingMC. Note that our algorithm MedianOfMedians finds the optimal solution essentially by searching for the optimal marginal cost using $O(\log mn)$ calls to MatchingMC which itself costs $(m \log n)$ queries, and the query complexity upper bound is in fact the product of the two aforementioned lower bounds. This suggests that improving the query complexity upper bound further would require an algorithm that does not treat MatchingMC as a black-box. Due to space limitations, we defer the proofs in this section to Appendix C.

THEOREM 3. *Any algorithm needs $\Omega(m \log n)$ queries to find the optimal bidding strategy, even if it knows the value $k$ for which $\boldsymbol{\mu}^k$ is the almost-optimal integral bidding strategy.*

THEOREM 4. *Any algorithm needs $\Omega(\log(mn))$ queries to find the optimal bidding strategy, even if it has access to a black-box oracle of MatchingMC that uses a single query.*

## 6  Learning-Augmented Algorithms

In this section we aim to design searching algorithm that utilize a (possibility erroneous) prediction $\hat{\boldsymbol{\mu}}$ regarding the actual optimal fractional strategy $\boldsymbol{\mu}^o$. The error of the prediction is measured by its distance to the optimal solution in the $\ell$-infinity norm, i.e.

$$\eta = \max_j |\hat{\mu}_j - \mu_j^o|. \tag{6}$$

We show the following algorithm, modified from MedianOfMedians, achieves a query complexity of $O(m \log m\eta \log \eta)$, note that since $\eta \le n$, this guarantee matches the query complexity of MedianOfMedians even if the prediction is arbitrarily wrong.

The algorithm begins by checking whether the floor of the predicted bidding strategy, $\lfloor \hat{\mu}_j \rfloor$, for all $j$, is ALMOST-OPTIMAL using OptCheck. If it is, the algorithm applies RoundUp and returns the optimal solution. If not, the algorithm assumes the error is small and attempts to search for the optimal solution within a restricted range around the predicted strategy $\hat{\mu}_j$ on each platform, following a similar approach to the MedianOfMedians algorithm.

If the optimal solution is still not found, the search range is expanded, and the search is repeated. This process continues until a almost-optimal solution is identified. By progressively expanding the search range as the square of the previous range, we show that the query complexity is at most $O(m \log(m\eta) \log \eta)$. Please refer to Algorithm 2 for a formal description.

---

**ALGORITHM 2:** BranchOutMedianOfMedians

1  **Initialize:** $\ell_j \leftarrow \hat{\mu}_j, r_j \leftarrow \hat{\mu}_j$ **for all** $j \in M$
2  $\hat{\boldsymbol{\mu}} \leftarrow \lfloor \hat{\boldsymbol{\mu}} \rfloor$
3  **if** OptCheck($\hat{\boldsymbol{\mu}}$) == ALMOST-OPTIMAL **then return** RoundUp($\hat{\boldsymbol{\mu}}$)
4  $i \leftarrow 0$              // initialize the counter for doubling process
5  **while** TRUE **do**
6      $\ell_j \leftarrow \hat{\mu}_j - 2^{2^i}, r_j \leftarrow \hat{\mu}_j + 2^{2^i}$ for all $j \in M$
7      range-indicator $\leftarrow$ TRUE      // assume range is correct
8      **while** range-indicator == TRUE **do**
9          $\mu_j \leftarrow \frac{\ell_j + r_j}{2}$ for all $j \in M$
10         query each platform $j$ strategy $\mu_j$, obtain $v_j(\mu_j)$, $c_j(\mu_j)$ and $MC_j(\mu_j)$
11         rank the platforms in non-decreasing order of $\mu_j$ s.t. if $i \le j$, $\mu_i \le \mu_j$
12         $j^* \leftarrow \min_j(|\sum_{i \le j}(r_i - \ell_i) - \sum_{i \ge j}(r_i - \ell_i)|)$  // find the $j^*$ that equally split the search space
13         $k \leftarrow MC_{j^*}(\mu_{j^*})$
14
15         **if** there exist a $MC_j(\hat{\mu}_j - 2^{2^i}) > k$ **then**
16             $\ell_j \leftarrow \mu_j + 1$ for $j \le j^*$          // $k$ is too small
17
18         **else**
19             $\boldsymbol{\mu}^k \leftarrow$ MatchingMC($[\hat{\mu}_j - 2^{2^i}, \hat{\mu}_j + 2^{2^i}] \forall j, k$)
20             **if** OptCheck($\boldsymbol{\mu}^k$)) == NOT-$\boldsymbol{\mu}^k$ **then**
21                 $r_j \leftarrow \mu_j - 1$ for all $j \ge j^*$   // $k$ is too large
22
23             **else if** OptCheck($\boldsymbol{\mu}^k$) == INFEASIBLE **then**
24                 $r_j \leftarrow \mu_j - 1$ for all $j \ge j^*$   // $k$ is too large
25
26             **else if** OptCheck($\boldsymbol{\mu}^k$) == ALMOST-OPTIMAL **then**
27                 **return** RoundUp($\boldsymbol{\mu}^k$)
28
29             **else**
                // OptCheck($\boldsymbol{\mu}^k$) == NOT-OPTIMAL
30                 $\ell_j \leftarrow \mu_j + 1$ for all $j \le j^*$   // $k$ is too small
31
32         **if** there exist a platform with $\ell_j > r_j$ **then**
33             range-indicator $\leftarrow$ FALSE // search in the given range is complete
34     $i \leftarrow i + 1$                // update the search range

---

THEOREM 5. *Given any instance $\mathcal{I}$, and predicted optimal bidding strategy $\hat{\boldsymbol{\mu}}$ such that the error of the $\hat{\boldsymbol{\mu}}$ is $\eta$, the BRANCHOUTMEDIANOF-MEDIANS algorithm finds the optimal bidding strategy with at most $O(m \log m\eta \log \eta)$ queries.*

PROOF. We first argue the correctness of the algorithm. Let $\eta$ denote the error of the prediction as defined in (6), and let $i^*$ be the smallest $i$ such that $2^{2^i} \geq (\eta + 1)$. We will show that the algorithm does not terminate in any round $i < i^*$. Let $\boldsymbol{\mu}^*$ represent the almost-optimal integral bidding strategy. When $i < i^*$, there exists at least one platform $j$ such that $\mu_j^* \notin [\ell_j, r_j]$ at the beginning of the $i^*$-th iteration of the outer while loop. By the correctness of OPTCHECK and since the algorithm only searches within the range $[\ell_j, r_j]$, it cannot return a solution in earlier rounds.

Next, we argue that the algorithm will terminate in round $i^*$ with the optimal solution. Since $2^{2^{i^*}} \geq (\eta + 1)$, we know that $\mu_j^* \in [\ell_j, r_j]$ for all $j$ at the start of the $i^*$-th iteration of the outer while loop. Now, consider the search process during this round. As in the proof of Theorem 2, we show correctness by proving that during the execution of the $i^*$-th iteration, there always exists some $\mu \in [\ell_j, r_j]$ such that $\mu^{\text{MC}_j(\mu)} = \boldsymbol{\mu}^*$. Considering all possible updates to $\ell_j$ and $r_j$ for each platform, we will now show that none of these updates eliminate any such $\mu$ values.

First, in Line 16, the algorithm encounters a platform $j$ where $\text{MC}_j(\hat{\mu}_j - 2^{2^{i^*}}) > k$, meaning the current candidate marginal cost $k$ is smaller than the marginal cost of the smallest strategy within the current search range for that platform. Since $\mu_j^* \in [\hat{\mu}_j - 2^{2^{i^*}}, \hat{\mu}_j + 2^{2^{i^*}}]$ for all platforms and $\boldsymbol{\mu}^* = \boldsymbol{\mu}^k$ for some $k$, this implies that the current marginal cost candidate $k$, as well as all marginal costs weakly smaller than $k$, cannot correspond to the optimal marginal cost $\boldsymbol{\mu}^*$. These marginal costs (and their corresponding strategies) are thus eliminated from the search range.

In Line 21, the algorithm is in the case where $\text{OPTCHECK}(\boldsymbol{\mu}^k) == \text{NOT} - \boldsymbol{\mu}^k$, indicating that $\boldsymbol{\mu}^k$ is not optimal. This implies that there exists a platform $j$ such that: 1. $\mu_j^k = \hat{\mu}_j + 2^{2^{i^*}}$, i.e., the largest strategy, and 2. for the same platform $j$, $\text{MC}_j(\hat{\mu}_j + 2^{2^{i^*}} + 1) \leq k$. This means that $k$, along with all marginal costs weakly greater than $k$, exceeds the optimal marginal cost corresponding to $\boldsymbol{\mu}^*$. These marginal costs (and their corresponding strategies) are therefore eliminated from the search range.

The remaining cases are handled in the same way as discussed in Theorem 2. In Line 24, when $\boldsymbol{\mu}^k$ is infeasible, we eliminate all marginal costs weakly greater than the current one being tested. In Line 30, when $\boldsymbol{\mu}^k$ is not optimal, we eliminate all marginal costs weakly smaller than the current one. Finally, in Line 27, once we find $\boldsymbol{\mu}^*$, we use ROUNDUP to obtain the optimal fractional strategy.

We now prove the query complexity of the algorithm. Let $i^*$ be the value of $i$ when the algorithm terminates. First, we have $i^* \leq \eta^2$, where $\eta$ is the error of the prediction as defined in (6). By Lemma 4, we know that the MATCHINGMC operation in iteration $i$ takes $m \log 2^{2^i}$ time. Additionally, by Theorem 2, the while loop within this iteration will run $\log(m 2^{2^i})$ times. Since all other subroutines take $O(m)$ queries, and the size of the search range is squared at each step, the algorithm terminates when the search

space is weakly larger than $n$.

$$\sum_{i=0}^{\log \log i^*} m \log(m \cdot 2^{2^i}) \cdot \log 2^{2^i}$$

$$= \sum_{i=0}^{\log \log i^*} m(\log m + 2^i) 2^i$$

$$= m(\log m + 2^{\log \log i^* + 1}) \cdot 2^{\log \log i^* + 1}$$

$$= m(\log m + 2 \log i^*) \cdot 2 \log i^*$$

$$\leq m(\log m + 2 \log(\eta)^2) \cdot 2 \log(\eta)^2$$

$$= m(\log m + 4 \log(\eta)) \cdot 2 \log(\eta)$$

$$= O(m \log(m\eta) \log \eta) \qquad \square$$

As a corollary, we also achieved "best-of-both-worlds" results in terms of consistency and robustness. Specifically, if the provided prediction is correct (or even "almost correct," i.e., $\lfloor \hat{\boldsymbol{\mu}} \rfloor = \lfloor \boldsymbol{\mu}^o \rfloor$), only $2m$ queries are required (note that even checking that a bidding profile is feasible requires $m$ queries). Since $\eta \leq n$ by definition, the total number of queries will never exceed $O(m \log(mn) \log n)$, which matches the query complexity of MEDIANOFMEDIANS. Formally,

COROLLARY 1. *The BRANCHOUTMEDIANOFMEDIANS algorithm is $2m$-consistent and $O(m \log mn \log n)$ robust, where the robustness matches the query complexity of MEDIANOFMEDIANS.*

## 7 Conclusion

In this work, we addressed the challenge of finding the optimal bidding strategy for advertisers operating in a multi-platform auction environment with low query complexity. Our approach models competition within each platform through value and cost functions that map various bidding strategies to their respective outcomes. We introduced an efficient algorithm that achieves this goal with a query complexity of $O(m \log(mn) \log n)$, where $m$ represents the number of platforms and $n$ denotes the number of potential bidding strategies available on each platform.

To further enhance efficiency, we incorporated the learning-augmented framework, proposing an algorithm that leverages a potentially flawed prediction of the optimal bidding strategy. Our results provide a query complexity bound that degrades gracefully, achieving $O(m)$ queries when accurate predictions are available and $O(m \log(mn) \log n)$ even with completely incorrect predictions. This flexibility exemplifies a "best-of-both-world" scenario, providing advertisers with different options to effectively navigate the complexities of multi-platform bidding with minimal queries.

We believe that autobidding in multi-platform auction settings is understudied, and many intriguing questions remain unanswered. One immediate question for exploration is closing the gap between the upper and lower bounds established in our work, which would necessitate different tools and ideas. Additionally, it would be interesting to analyze the dynamics of the market if all the bidders adopted the approach presented in this study in determining their bidding strategies.

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

# A  Proof of Lemma 2

PROOF. For presentation purpose, we drop the subscript $j$ in this prove as it should hold for any platform. We define $c(0)/v(0) = 0$ We prove the statement via induction. First consider the base case for $\mu = 1$, we have $c(1)/v(1) \geq 0 = c(0)/v(0)$ since both the cost and the value functions weakly increase w.r.t $\mu$, we also have $c(1)/v(1) = \frac{c(1)-c(0)}{v(1)-v(0)} = \mathrm{MC}(1)$ by definition. The base case is therefore established.

Let $\frac{c(\mu)}{v(\mu)} = X_\mu$. Assume, for induction, that $X_{\mu'} \leq \mathrm{MC}(\mu')$ for any $\mu' < \mu$. We first show $X_{\mu-1} \leq X_\mu$ holds for $\mu \geq 2$. Consider

$$c(\mu) = X_\mu \cdot v(\mu)$$
$$c(\mu) - c(\mu-1) = X_\mu \cdot v(\mu) - c(\mu-1)$$
$$\mathrm{MC}(\mu) \cdot (v(\mu) - v(\mu-1)) = X_\mu \cdot v(\mu) - c(\mu-1)$$
$$\mathrm{MC}(\mu-1) \cdot (v(\mu) - v(\mu-1)) = X_\mu \cdot v(\mu) - c(\mu-1)$$
$$X_{\mu-1} \cdot (v(\mu) - v(\mu-1)) \leq X_\mu \cdot v(\mu) - c(\mu-1)$$
$$X_{\mu-1} \cdot (v(\mu) - v(\mu-1)) \leq X_\mu \cdot v(\mu) - X_{\mu-1} \cdot v(\mu-1)$$
$$X_{\mu-1} \cdot v(\mu) \leq X_\mu \cdot v(\mu)$$
$$X_{\mu-1} \leq X_\mu,$$

where the third equality is by definition of $\mathrm{MC}$, the forth equality is by monotoncity of $\mathrm{MC}$, and the first inequality is by induction hypothesis $\mathrm{MC}(\mu-1) \geq X_{\mu-1}$. In addition, consider the same set of equation again:

$$c(\mu) = X_\mu \cdot v(\mu)$$
$$c(\mu) - c(\mu-1) = X_\mu \cdot v(\mu) - c(\mu-1)$$

$$c(\mu) - c(\mu - 1) = X_\mu \cdot v(\mu) - X_{\mu-1} \cdot v(\mu - 1)$$
$$c(\mu) - c(\mu - 1) \geq X_\mu \cdot v(\mu) - X_\mu \cdot v(\mu - 1)$$
$$\text{MC}(\mu) \cdot (v(\mu) - v(\mu - 1)) \geq X_\mu \cdot (v(\mu) - v(\mu - 1))$$
$$\text{MC}(\mu) \geq X_\mu$$

where as the first inequality is due to $X_{\mu-1} \leq X_\mu$, we therefore have $\text{MC}(\mu) \geq X_\mu$, hence proved.  □

## B  Missing Proofs from Section 4

### B.1  Proof of Lemma 4

Proof. The algorithm performs binary search on each platform to find the maximum $\mu$ such that $\text{MC}_j(\mu) \leq k$, since binary search checks $O(\log(r_j - \ell_j))$ number of choice of $\mu$ on each platform and there are $m$ platforms, the query complexity is $O(m \log \max_j(r_j - \ell_j))$.

We now prove the correctness of the algorithm via case analysis, i.e., for each platform $j$, we have $\mu_j^k = \max_\mu(\text{MC}_j(\mu) \leq k)$. Fix any arbitrary platform $j$, if $\text{MC}_j(r_j) \leq k$, the algorithm will keep update $\ell_j$ until eventually $\ell_j = r_j$ and correctly set $\mu_j^k = \ell_j = r_j$. On the other hand, if $\text{MC}_j(r_j) > k$, by the termination condition, we know that $\text{MC}_j(\mu_j^k + 1) > k$, $\text{MC}_j(\mu_j^k) \leq k$, which corresponds to $\mu_j^k$ being the maximum bid with a marginal cost weakly less than $k$.  □

### B.2  Proof of Lemma 5

Proof. Since OptCheck queries at most 2 strategies from each platform, the worst-case number of queries used is $2m = O(m)$. We now prove the correctness of the subroutine for each different case. The INFEASIBLE case is trivial. For the NOT-$\boldsymbol{\mu}^k$ case, as indicated by line 8, since there exists a platform where $\text{MC}_j(\mu_j + 1) \leq k$, we know that the given bidding profile $\boldsymbol{\mu}$ is not $\boldsymbol{\mu}^k$ for some $k$ by definition. On the other hand, if OptCheck does not terminate in line 9, it means $\boldsymbol{\mu}$ is feasible and $\boldsymbol{\mu} = \boldsymbol{\mu}^k$ for some $k$. To check if the given profile is almost-optimal (the floor of $\boldsymbol{\mu}^o$), by Lemma 3, we just need to verify whether increasing $k$ would make $\boldsymbol{\mu}^k$ infeasible. If the next immediate change would cause $\boldsymbol{\mu}$ to be infeasible, then $\boldsymbol{\mu}$ is ALMOST-OPTIMAL; otherwise, it is NOT-OPTIMAL.  □

## C  Missing proofs from section 5

### C.1  Proof of Theorem 3

Proof. Given any algorithm, assume it knows the correct value of $k$. On each platform, finding the maximum $\mu$ (therefore the $\lceil \mu_j^o \rceil$) such that $\text{MC}_j(\mu) \leq k$ takes at least $\Omega(\log n)$ queries. We prove this via a decision tree argument similar to the $\Omega(\log n)$ query complexity for the binary search problem.

Fix an arbitrary platform $j$; we want to determine the maximum index $\mu$ such that $\text{MC}_j(\mu) \leq k$. We represent any algorithm as a decision tree as follows:

(1). Each query made to the platform is represented as a node in the decision tree, and each node has three children: one for $\text{MC}_j(\mu) \leq k$, one for $\text{MC}_j(\mu) > k$, and a third for cases not specified. (2). The leaves of this tree represent the possible outcomes of the search: specifically, finding the maximum index $\mu$ such that $\text{MC}_j(\mu) \leq k$.

There are $n+1$ distinct outcomes, corresponding to the maximum value of $\mu$ being 0, 1, ..., or $n$. In any decision tree with $x$ leaves, the minimum height $h$ is $\log x$.

Moreover, the height $h$ of the decision tree corresponds to the number of queries made. Therefore, the minimum height of the decision tree is $\log(n+1)$, implying that the number of queries needed to resolve the search will be at least $\Omega(\log(n + 1)) = \Omega(\log n)$.

Lastly, since all platforms operate independently, the search on each platform requires $\Omega(\log n)$ queries. Consequently, to complete the search across all platforms will require $\Omega(m \log n)$ queries.  □

### C.2  Proof of Theorem 4

Proof. There are a total of $mn$ distinct marginal costs, and our objective is to determine the marginal cost $\text{MC}_j(\mu)$ for a specified $j$ and $\mu$ such that $\boldsymbol{\mu}^{\text{MC}_j(\mu)}$ represents the almost-optimal integral solution. We establish this by reducing the binary search problem to our problem.

Consider a binary search scenario involving a single sorted array $A$ with $|A| = h$ and a target number $a$ for which we are searching. Let $i$ denote the index of $a$ within this array. We can construct an instance of our problem featuring a global ordering of marginal costs. In this global ordering, the marginal costs $\text{MC}_j(\mu)$ located at index $i$ correspond to the bidding strategy $\boldsymbol{\mu}^{\text{MC}_j(\mu)}$, which serves as the almost-optimal integral solution. If we are able to identify the index of the optimal marginal cost in fewer than $\Omega(\log mn)$ queries, it would consequently allow us to resolve the binary search problem in fewer than $\Omega(\log h)$ queries. This outcome would contradict the established complexity bounds associated with binary search.  □

