# OpenReview forum: "Multi-Platform Autobidding with and without Predictions"
_ACM.org/TheWebConf/2025/Conference — WWW 2025 Poster_

### Official Review · Reviewer_2Ye4 · 2024-11-12

**Novelty:** 6
**Technical Quality:** 6

**Review:**

This paper studies how to maximize the total value by bidding on m different platforms, subject to a budget B and a return-on-spend (ROS) constraint.
The problem assumes that there are n different bidding strategies for each platform, each with a different cost and value.
The bidder does not know the quality of each strategy, and has to make a query to get such information.
The main goal is to find optimal strategies with the fewest queries.
The problem is somewhat like a knapsack problem with specific items and special ROS constraints.

The naive search space is n^m.
The authors exploits some specific structure in the strategies, that is, n strategies are ordered in their increasing costs and values.
They show that there exists a constraint on the marginal cost between two consecutive strategies, which when satisfied, reveals an “almost-optimal” solution.
This constraint reduces the search space to O(mn), so a proper binary search plus a quality checker leads to a much faster algorithm with O(m log(mn) logn) queries.
The authors also augments the algorithm with ML predictions, and show reliable improvements can be made when predictions are accurate.
I think it is a nice paper that has solid technical contributions.

There is room to be improved in the writing.

- In Sec. 1.1, the notation mu and the notion of n strategies are vague.

- Line 303: The error eta of an predictions

- In Example 1, please explain that inequalities are about ROS.

- Please remove blank lines in Alg. 2.

**Questions:**

I am confused when the authors say “starting with an initial budget of B, we allocate an infinitesimal amount to the platform currently offering the best value-to-unit-cost ratio (equivalently, the smallest marginal cost).”
The search is mainly binary (median) search, where does the infinitesimal amount come into play? Or the authors meant that this amount is needed to make every query. If so, please clarify this explicitly in the paper as an assumption.

**Reviewer Confidence:**

2: The reviewer is willing to defend the evaluation, but it is likely that the reviewer did not understand parts of the paper

**Scope:**

4: The work is relevant to the Web and to the track, and is of broad interest to the community

---

### Official Review · Reviewer_FGMc · 2024-11-19

**Novelty:** 3
**Technical Quality:** 4

**Review:**

Pros:

The paper examines the popular formulation of autobidding problems with budget and ROS restrictions.The problem is truly relevant and interesting to the community.

The article is based on a clear system of premises and includes a theoretical justification for the proposed algorithm.

 I assume that the theoretical part of the article devoted to the algorithm itself contains novelty and is carefully proven.

Cons:

I would like to see more traditional vocabulary in the article (for example, not a program, but a problem),fewer typos (missing and upper/lowercase letters). Furthermore, I'd like to find more elegant presentation of mathematical material (linear interpolation formula, the calculations in Lemma 1 and Example 1 seem redundant).

Moving on to more serious comments, I would like to note that monotonicity and, especially, convexity of marginal cost is not a standard assumption in auction theory, which significantly narrows the scope of applicability of the algorithm.

In the proof of Lemma 2  the forth equality has to be inequality.

As in any article on the topic of robustness and predictions, I would like to see in this article an empirical part with verification of the resulting algorithms using metrics corresponding to the formulation of the problem.

**Questions:**

Can the authors refer to works that indicate the undoubted convexity of MC in problems related to auctions?

**Reviewer Confidence:**

2: The reviewer is willing to defend the evaluation, but it is likely that the reviewer did not understand parts of the paper

**Scope:**

3: The work is somewhat relevant to the Web and to the track, and is of narrow interest to a sub-community

---

### Official Review · Reviewer_FSzY · 2024-11-30

**Novelty:** 3
**Technical Quality:** 3

**Review:**

This article proposes an optimal/sub-optimal strategy to the bidding for advertiser in a multi-platform auction setting. This research is well motivated and the application scenario is very interesting. The bidding strategies are proved with some analysis and investigated through a simple numerical example. The paper is well written; however, it is hard to follow.

The numerical example is rather simple and can’t really demonstrate the utility of the proposed strategies. Hence it is hard to evaluate the significance of this research.

**Questions:**

1. what is the complexity of implementing such biding strategies in practice?
2. is there any barrier of proving the bidding optimality?

**Reviewer Confidence:**

1: The reviewer's evaluation is an educated guess

**Scope:**

3: The work is somewhat relevant to the Web and to the track, and is of narrow interest to a sub-community

---

### Official Review · Reviewer_zije · 2024-12-02

**Novelty:** 3
**Technical Quality:** 3

**Review:**

This paper studies how to help advertisers find optimal bidding strategies across multiple advertising platforms when they have limited information and need to learn through queries. They propose two algorithms - a basic one and an enhanced version using machine learning predictions - that significantly reduce the number of queries needed to find the optimal bidding strategy, achieving O(m) queries in the best case while maintaining robustness against poor predictions.

pros:

1.Clear problem formulation and assumptions

2.Comprehensive lower bound analysis

3.Strong theoretical guarantees

cons:

1.Lacks empirical validation

2.Some practical assumptions may be oversimplified

3.Implementation details for practitioners could be more detailed

**Questions:**

1. Could you provide empirical validation?

2. Can it be extended to more complex bidding scenarios?

3. Does the diminishing returns assumption always hold in reality?

**Reviewer Confidence:**

1: The reviewer's evaluation is an educated guess

**Scope:**

3: The work is somewhat relevant to the Web and to the track, and is of narrow interest to a sub-community

---

### Official Review · Reviewer_9s56 · 2024-12-03

**Novelty:** 5
**Technical Quality:** 5

**Review:**

This work addresses the challenging problem of optimizing bidding strategies across multiple auction platforms, presenting innovative algorithms with robust theoretical guarantees. The clarity of the paper is commendable, with well-structured explanations and logical flow. Its originality lies in adopting a learning-augmented framework to integrate predictions into bidding optimization, showcasing a significant improvement in query efficiency. Furthermore, the significance of the contribution is underscored by its practical relevance for advertisers and its theoretical implications for multi-platform optimization. However, there are a few areas for improvement. The practical applicability of the proposed methods may require further validation through real-world experiments or simulations, as the current evaluation is predominantly theoretical. Additionally, the paper could benefit from deeper discussions on computational overheads and scalability to extremely large platforms. Overall, the work provides a strong foundation for advancing autobidding strategies while highlighting directions for future exploration.

**Questions:**

1. How do the algorithms perform in scenarios with a very large number of platforms or highly granular bidding strategies? Are there any computational bottlenecks?
2. Can you provide additional insights or examples illustrating how the algorithms handle extreme cases of prediction errors, especially in adversarial settings?
3. Have the proposed algorithms been tested in real-world scenarios or simulated environments to validate their effectiveness and efficiency beyond theoretical bounds?

**Reviewer Confidence:**

2: The reviewer is willing to defend the evaluation, but it is likely that the reviewer did not understand parts of the paper

**Scope:**

3: The work is somewhat relevant to the Web and to the track, and is of narrow interest to a sub-community